# AΚtransU-Net: Transformer-Equipped U-Net Model for Improved Actinic Keratosis Detection in Clinical Photography

**DOI:** 10.3390/diagnostics15141752

**Published:** 2025-07-10

**Authors:** Panagiotis Derekas, Charalampos Theodoridis, Aristidis Likas, Ioannis Bassukas, Georgios Gaitanis, Athanasia Zampeta, Despina Exadaktylou, Panagiota Spyridonos

**Affiliations:** 1Department of Medical Physics, Faculty of Medicine, School of Health Sciences, University of Ioannina, 45110 Ioannina, Greece; p.derekas@uoi.gr; 2Department of Computer Science & Engineering, School of Engineering, University of Ioannina, 45110 Ioannina, Greece; theodoridisxaris01@gmail.com (C.T.); arly@uoi.gr (A.L.); 3Department of Skin and Venereal Diseases, Faculty of Medicine, School of Health Sciences, University of Ioannina, 45110 Ioannina, Greece; ibassuka@uoi.gr (I.B.); ggaitan@uoi.gr (G.G.); athanasiazampeta@gmail.com (A.Z.); 4Department of Dermatology, General Hospital of Nikaia—Piraeus “Agios Panteleimon”, 18454 Nikaia, Greece; deppie@gmail.com

**Keywords:** medical image segmentation, U-net, transformer, skin lesions, actinic keratosis, cutaneous cancerization field, AI in dermatology

## Abstract

**Background:** Integrating artificial intelligence into clinical photography offers great potential for monitoring skin conditions such as actinic keratosis (AK) and skin field cancerization. Identifying the extent of AK lesions often requires more than analyzing lesion morphology—it also depends on contextual cues, such as surrounding photodamage. This highlights the need for models that can combine fine-grained local features with a comprehensive global view. **Methods:** To address this challenge, we propose AKTransU-net, a hybrid U-net-based architecture. The model incorporates Transformer blocks to enrich feature representations, which are passed through ConvLSTM modules within the skip connections. This configuration allows the network to maintain semantic coherence and spatial continuity in AK detection. This global awareness is critical when applying the model to whole-image detection via tile-based processing, where continuity across tile boundaries is essential for accurate and reliable lesion segmentation. **Results:** The effectiveness of AKTransU-net was demonstrated through comparative evaluations with state-of-the-art segmentation models. A proprietary annotated dataset of 569 clinical photographs from 115 patients with actinic keratosis was used to train and evaluate the models. From each photograph, crops of 512 × 512 pixels were extracted using translation lesion boxes that encompassed lesions in different positions and captured different contexts. AKtransU-net exhibited a more robust context awareness and achieved a median Dice score of 65.13%, demonstrating significant progress in whole-image assessments. **Conclusions:** Transformer-driven context modeling offers a promising approach for robust AK lesion monitoring, supporting its application in real-world clinical settings where accurate, context-aware analysis is crucial for managing skin field cancerization.

## 1. Introduction

Actinic keratoses (AKs) are common precancerous skin lesions that develop in areas of chronic ultraviolet radiation exposure. They are considered early indicators of squamous cell carcinoma [1] and often coexist with subclinical changes in the surrounding skin, highlighting their key role in the broader process of skin field cancerization. Managing AKs effectively requires addressing both individual lesions and the entire affected skin field to reduce the risk of progression to invasive skin cancers [2].

The clinical evaluation of AK involves assessing the extent of the AK lesions (the AK burden), which is crucial for determining baseline disease severity and guiding treatment decisions.

In routine clinical practice, assessing the AK burden presents challenges. Most guidelines rely on a lesion count, yet specialists often find this metric inconsistent [3]. Additionally, studies indicate that complete patient clearance rates inversely correlate with baseline lesion numbers, yet many intervention studies fail to account for this baseline burden, potentially underestimating treatment’s benefits [4,5].

An accurate evaluation of treatment outcomes and long-term monitoring are critical to optimizing patient care. Current methods for assessing the AK burden and treatment response, such as clinical scoring systems and qualitative indices, are subjective and prone to interobserver variability [6,7]. These limitations underscore the need for more objective, reproducible, and comprehensive approaches that can enhance management strategies and improve the surveillance of AK over time.

Clinical photography has become an essential documentation tool in dermatology, particularly for diagnosing and monitoring conditions such as actinic keratoses (AKs). It allows clinicians to track changes in skin lesions over time, particularly in cases of extensive sun damage where AKs appear diffusely across large body surface areas [8]. Clinical photography, coupled with image analysis, offers an efficient means of the comprehensive documentation of AK clearance and/or the formation of new AKs.

Detecting AK lesions in clinical photographs poses a significant challenge due to their subtle visual characteristics and ill-defined boundaries. Their variability in size, shape, color, and texture further complicates the automated detection process. Early attempts at quantitatively assessing AK lesions using clinical photography faced challenges such as inconsistent illumination, lesion color diversity [9], and the restriction of detection to preselected subregions of the photographed skin areas using a binary patch classifier for the discrimination of Aks from healthy skin [10,11,12]. These limitations hindered the reproducibility and accuracy of detection.

Leveraging an optimized actinic keratosis convolutional neural network (AKCNN) as a patch classifier has significantly improved the detection of AKs [13]. However, a patch classifier processes small image patches (e.g., 50 × 50 pixels) independently, classifying each patch without explicit consideration of its surrounding spatial context. An AKCNN is subject to manually predefined scanning areas necessary to exclude skin regions prone to false diagnoses, such as actinic keratosis, on clinical images. Unlike more distinct skin conditions, AK lesions often blend with the surrounding sun-damaged skin, making it challenging to delineate them precisely. These limitations prompted our earlier work, which explored semantic segmentation using a U-net model with recurrent spatial processing in skip connections, thereby enhancing the detection of AKs using clinical photographs (AKU-net) [14]. 

However, an accurate identification of the AK burden within skin field cancerization often relies more on contextual cues—such as surrounding photodamage—than on the lesion’s appearance alone. This dependence on broader spatial information underscores the importance of models that integrate fine-grained local features and a global context to effectively detect and segment AK lesions in a real clinical photography setting. 

Moreover, the high-resolution nature of the clinical photographs and the need to detect fine-texture AK regions impose additional limitations when implementing a regular semantic segmentation method. Training a segmentation network on high-resolution images requires either a huge memory capacity, which is often unavailable, or progressively reducing the image’s spatial size to fit into the memory. Nevertheless, down-sampling increases the risk of losing critical discriminative features. To address the challenge of segmenting AK lesions from clinical images, we have previously adopted the “tile strategy” [14]. This method involves breaking down a large image into smaller tiles (crops), making predictions on these tiles, and then stitching them back together to form the final prediction. However, this process may result in the loss of image features shared between tiles, leading to under-segmentation or the missed detection of AK lesions.

As semantic segmentation evolves rapidly, it is essential to continually reassess and refine architectural choices to meet the growing demands of complex clinical tasks, including the evaluation of the AK burden and longitudinal monitoring in dermatology. Within this context, the main contributions of our study are as follows:We explore architectural U-based enhancements that integrate Transformers to enable more coherent and contextually informed AK segmentations.We introduce AKTransU-net, a hybrid U-net-based architecture that incorporates Transformer blocks to enrich feature representations. These are further refined through ConvLSTM modules embedded within the skip connections. Improvements in AK segmentation accuracy—at both crop and full-image levels—are demonstrated through comparisons with state-of-the-art semantic segmentation models.We significantly improve the whole-image segmentation performance by employing a Gaussian weighting technique to mitigate boundary effects during prediction merging.We underscore the crucial role of scanning methodology in real-world segmentation applications, suggesting that evaluations conducted solely under ideal conditions may fail to capture a model’s variability and robustness during deployment.

The remainder of the paper is organized as follows: Section 2 reviews related work on Transformer architectures in medical image segmentation, with an emphasis on hybrid U-net-based models and enhancements to their skip connections. Section 3 introduces the proposed AKtransU-net architecture, describes the materials, the data preparation, and details the evaluation methodology. Section 4 presents comparative results on both localized image crops and full-image regions. Section 5 discusses the main findings, highlights the study’s limitations, and outlines future research directions. Finally, Section 6 concludes the paper.

## 2. Related Work

### 2.1. The Evolution of Transformer Architectures in Medical Image Segmentation

Initially developed for natural language processing, Transformers have increasingly demonstrated strong potential in computer vision. The introduction of the Vision Transformer (ViT) by Dosovitskiy et al. [15] marked a significant shift, establishing the viability of Transformers for visual tasks. Unlike traditional convolutional neural networks (CNNs), Transformers leverage self-attention mechanisms to model long-range dependencies across the entire image, enabling a deeper understanding of spatial relationships and context. Although ViT was developed for natural image understanding, it quickly attracted attention for dense prediction tasks such as semantic segmentation due to its powerful global context modeling. However, early efforts to directly apply pure Transformer architectures to pixel-wise segmentation encountered challenges, including a high computational cost and limited spatial precision. To mitigate these issues, various Transformer variants were introduced. The Swin Transformer [16] improved computational efficiency and spatial locality by computing self-attention within shifted windows. Other variants, such as Pyramid Vision Transformer (PVT) [17] and LeViT [18], further optimized the balance between resolution and context modeling. These advances expanded the feasibility of integrating Transformers into medical segmentation tasks, paving the way for more practical and effective designs.

Since its introduction, U-net has inspired numerous enhanced variants, demonstrating a widespread adoption in medical image segmentation [19,20]. The rise of Transformer architectures has further driven the development of hybrid models that combine U-net’s strengths with global attention mechanisms. The review papers in the bibliography reflect this rapid evolution and highlight the growing impact of these hybrid approaches in the medical field [21,22,23].

One of the earliest and most influential examples is TransU-net, which integrates a ViT module into the U-net encoder [24]. In this architecture, a CNN backbone is first employed for local feature extraction, followed by the ViT component, which captures long-range dependencies across the spatial domain. More recently, the TransU-net framework has been extended to include Transformer modules that can be flexibly inserted into the U-net backbone, resulting in three hybrid configurations: Encoder-only, Decoder-only, and Encoder + Decoder. This modular design allows users to tailor the architecture more easily to specific segmentation tasks [25].

Following TransU-net, several variants explored different ways to integrate Transformer blocks. Swin-Unet [26] replaced ViT with Swin Transformer blocks to enable efficient, hierarchical, and windowed attention across scales. DS-TransUnet [27] extended this concept by utilizing Swin Transformers in both the encoder and decoder paths, with dual-branch interactions to enhance its multi-scale consistency. UCTransnet [28], in contrast, focused on the skip connections, replacing them with a channel transposition and attention fusion mechanism.

Most recently, HmsU-net [29] was proposed as a multi-scale U-net architecture that integrates CNN and Transformer blocks in parallel at both the encoder and decoder. Each CNN-Trans block processes features through two branches, one convolutional and one Transformer-based, and fuses them using a multi-scale feature fusion module. A cross-attention mechanism further enhances skip connections across levels, enabling a stronger interaction between encoder and decoder representations. This design allows the network to effectively capture both local textures and global semantics [29].

These models illustrate the diversity of Transformer integration strategies within the U-net framework, ranging from encoder-only attention to full dual-path designs. Each variant reflects a unique attempt to balance spatial accuracy, global reasoning, and computational efficiency, making Transformer-enhanced U-nets a powerful class of models for the segmentation of medical images.

### 2.2. Skip Connections in Hybrid U-Net Architectures

Skip connections have been a cornerstone of U-net-based architectures, effectively bridging the encoder and decoder by preserving spatial details [30]. However, several studies have shown that plain skip connections, such as direct concatenation, can introduce semantic gaps between encoder and decoder features, negatively impacting segmentation performance. To address this, various models have rethought the design of skip connections. MultiResUnet [31], for example, instead of simply concatenating the feature maps from the encoder stages to the decoder stages, first passes them through a chain of convolutional layers with residual connections and then concatenates them with the decoder features. In another architecture, IBA-U-net [32], the researchers, to solve the difference between the feature map extracted from the encoding path and the output feature map of the upper layer in the decoding path, introduced the Attentive Bidirectional Convolutional Long Short-Term Memory block (BConvLSTM), which uses the BConvLSTM block to extract bidirectional information and then uses the attention block to compare BConvLSTM output elements to different degrees to highlight the salient features in the skip connection. UCTransnet [28] and FAFS-Unet [33] have introduced cross-attention and feature selection modules to enrich skip connections with context-aware filtering and semantic alignment, thereby enhancing their capabilities. Similarly, HmsU-net [29] improved the communication between the encoder and decoder by introducing a cross-attention module that fuses features from the last three encoder stages and distributes them to the decoder.

## 3. Materials and Methods

### 3.1. AKtransU-Net: Skip Connection Enforcement with Global Dependencies

Inspired by recent developments in hybrid U-net architectures and based on our previous exploration of U-net variants for the detection of AKs [14], the proposed architecture adopts the view that skip connections should actively propagate feature maps enriched with global dependencies to the decoder stage. To achieve this, we apply ViT blocks [15] to the encoder-derived feature maps at multiple scales. Passing the CNN feature maps through a Transformer block augments them with global relationships.

The output of each Transformer block is then passed into ConvLSTM modules within the skip connections, which process three inputs sequentially: the CNN encoder output, the Transformer features, and the up-sampled decoder output. This design enables the network to detect lesions not only by analyzing local textures but also by leveraging their broader spatial context. The architecture is shown in Figure 1.

#### 3.1.1. Transformer Encoder

In AKTransU-net, three Transformer encoder blocks are integrated at progressively down-sampled scales within the encoding path. This design enhances the CNN-extracted features with global attention, enabling the model to capture long-range dependencies across the spatial domain.

Each Transformer encoder block follows a standard ViT configuration (Figure 2), utilizing 16 attention heads (h = 16), and processes embedded tokens of size D = 1024. The encoder block consists of two sub-layers, each preceded by Layer Normalization and followed by a residual connection.

In the first sub-layer, the embedded patches are projected into sets of query (*Q*), key (*K*), and value (*V*) vectors through Multi-Head Self-Attention. For each head, attention is computed using scaled dot-product attention:(1)$$\mathrm{Attention}(Q,K,V)=softmax\left.\left(\;\frac{QK^{T}\;}{\sqrt{d_{k}\;}}\right.\right)V$$
where $d_{k}$ is the dimensionality of the key vectors calculated as follows:(2)$$d_{k}=\frac{D}{h}=\frac{1024}{16}=64$$

The outputs from all the attention heads are concatenated and passed through a linear projection. This is followed by a second Layer Normalization and a Feed-Forward Network composed of two linear layers with a non-linear activation function (Rectified Linear Unit). Finally, the output token sequence is reshaped back to its original spatial dimensions. This step ensures that the global context learned through attention is preserved while maintaining compatibility with the subsequent ConvLSTM and decoder layers.

#### 3.1.2. ConvLSTM in Skip Connections for Spatial Refinement

ConvLSTM layers are a variation of LSTMs (Long Short-Term Memory networks) that extend the traditional LSTM by incorporating convolutional operations, making them well-suited for capturing spatial relationships in image data [34,35,36,37]. While ConvLSTMs are often used in spatiotemporal tasks, they can be effectively applied to purely spatial data by focusing on spatial interactions across feature maps. ConvLSTM modules have been integrated into the U-net skip connections to enhance the model’s capacity to reconstruct accurate segmentation maps [32,37,38]

A ConvLSTM consists of an input gate $i_{t}$, an output gate $o_{t}$, a forget gate $f_{t}$, a memory cell $c_{t}$, and the hidden state $H_{t}$. The gates control access to, update, and clear the memory cell.

For an input $X_{t}$ the ConvLSTM process is formulated as follows:(3)$${\;i}_{t}=\sigma \left(W_{xi}*X_{t}+W_{hi}*H_{t-1}+W_{ci}\cdot c_{t-1}+b_{i}\right)$$(4)$${\;f}_{t}=\sigma \left(W_{xf}*X_{t}+W_{hf}*H_{t-1}+W_{cf}\cdot c_{t-1}+b_{f}\right)$$(5)$$c_{t}=f_{t}\cdot c_{t-1}+i_{t}\cdot \tanh\left(W_{xc}*X_{t}+W_{hc}*H_{t-1}+b_{c}\right)$$(6)$${\;o}_{t}=\sigma \left(W_{xo}*X_{t}+W_{ho}*H_{t-1}+W_{co}\cdot c_{t}+b_{o}\right)$$(7)$$H_{t}=o_{t}\cdot tanh(c_{t})$$
In the final step *t*, the refined feature map $\;H_{t}$ capturing spatial dependencies is passed to the decoder. $W_{x\cdot }$ and $W_{h\cdot }$ correspond to the 2D convolution kernel of the input and hidden states, respectively. (*) represents the convolution operation and a bullet (·) the Hadamard function (element-wise multiplication). $b_{i},\;b_{f},\;b_{c},\;and\;b_{o}$ are the bias terms, and  σ is the sigmoid function.

In our architecture, the ConvLSTM layer processes the CNN encoding, the Transformer encoding, and the up-sampled decoder feature map in a sequential three-step process. In the first step, the encoder’s CNN-based feature map is processed. In the second step, the Transformer-based representation is incorporated, using the hidden state from the previous step. In the third and final step, the up-sampled decoder feature map is refined, leveraging the accumulated spatial memory from the earlier steps. This sequential integration allows the ConvLSTM to serve as a spatial refinement module, enhancing the consistency of skip connections. As a result, the decoder receives a richer combination of low- and high-level spatial information, leading to more precise and accurate segmentation.

### 3.2. Material

The use of archival photographic material for this study was approved by the Human Investigation Committee (IRB) of the University Hospital of Ioannina (Approval No.: 3/17-2-2015 [θ.17]). The study included 115 patients diagnosed with facial AK, comprising 60 males and 55 females, aged between 45 and 85 years, who attended the specialized Dermato-Oncology Clinic of the Dermatology Department.

The inclusion criteria for the clinical photographs were the presence of clinically confirmed AKs depicted in the acquired and utilized photographs. The exclusion criteria were the existence of a biopsy-confirmed keratinocytic tumor (basal cell or squamous cell carcinoma). 

Facial photographs were captured with the camera positioned perpendicular to the target area, ensuring full coverage of the face from the chin to the hairline. High-resolution digital images (4016 × 6016 pixels) were obtained using a Nikon D610 (Nikon, Tokyo, Japan) equipped with a Nikon NIKKOR© 60 mm f/2.8G ED Micro lens, following a protocol adapted from Muccini et al. [38]. The camera was configured with an aperture of f/18, shutter speed of 1/80 s, ISO 400 [39], autofocus enabled, and white balance set to auto mode. A Sigma ring flash (Sigma, Fukushima, Japan) in TTL mode was mounted on the camera. Linear polarizing filters were placed in front of both the lens and the flash, aligned to create a 90° polarization angle between them, thereby minimizing surface reflections and enhancing the lesions’ visibility.

Two dermatologists jointly reviewed and annotated the photographs, reaching a consensus on the regions affected by AK.

#### Data Augmentation and Dataset Preparation

Semantic segmentation is a dense prediction task that requires pixel-level annotation. In medical imaging, however, obtaining such labels is particularly challenging, as it demands expert knowledge and is both time-consuming and costly.

Despite this, successful models have been trained on relatively small datasets. The solution to the limited availability of annotated medical images is excessive data augmentation [30].

While traditional augmentation techniques—such as elastic deformations, rotations, and flips—are commonly used to expand training data, we adopted a strategy more aligned with the clinical characteristics of our dataset. Specifically, multiple photographs were taken per patient to capture the presence of lesions across the entire face. These photographs were intended to provide different views of the same lesions, resulting in 569 annotated clinical photographs.

Given the high spatial resolution of the images (4016 × 6016 pixels), model training was performed using rectangular crops (tiles). The crop size was chosen to preserve sufficient contextual information. To this end, the images were initially downscaled by a factor of 0.5, and 512 × 512-pixel crops were extracted using translation-based lesion bounding boxes. This approach ensured that the resulting image tiles included lesions in varied positions and captured different contexts of the perilesional skin. In total, 16,891 lesion-center and translation-augmented patches were extracted. These augmentations provided natural variations in appearance and positioning without distorting the lesion morphology, allowing the model to learn more realistic and generalizable features.

For training, we used 510 clinical photographs acquired from 98 patients, from which 16,488 translation-augmented image crops were extracted. Approximately 20% of these crops (3298), corresponding to 15 patients, were reserved for validation. To evaluate the model’s performance, an independent test set comprising 59 photographs from 17 patients was used, yielding 403 central lesion crops without translation augmentation (Table 1).

### 3.3. Evaluation

To demonstrate the anticipated improvements in AKtransU-net in detecting AK lesions, we present comparative results using both CNN-based and CNN–Transformer hybrid architectures at the crop level. One of the CNN models evaluated was a deeper variant of AKU-net [14], which extends the original four hierarchical levels of AKU-net to five (AKU-net5). AKU-net is a U-net architecture with ConvLSTM modules integrated into the skip connections, without any Transformer components. Additionally, we assessed DeepLabv3+, a widely recognized benchmark model for the segmentation of medical images [40,41]. For CNN–Transformer hybrids, we included both TransU-net and HmsU-net in our comparisons. TransU-net is among the earliest and most extensively evaluated Transformer-based architectures for complex medical image segmentation tasks [24]. HmsU-net, currently considered a state-of-the-art model, has demonstrated superior performance across various medical image segmentation applications [29].

In semantic segmentation, several key evaluation metrics are employed to assess a model’s performance:(8)$$Precision\;=\;\frac{TP}{TP\;+\;FP}$$(9)$$Sensitivity\;=\;\frac{TP}{TP\;+\;FN}$$(10)$$Specificity\;=\;\frac{TN}{TN\;+\;FP}$$(11)$$Accuracy\;=\;\frac{TN\;+\;TP}{TN\;+\;FN\;+\;TP\;+\;FP}$$(12)$$Jaccard\;index\;=\;\frac{\left|A\cap B\right|}{\left|A\cup B\right|}=\frac{TP}{TP\;+\;FN\;+\;FP}$$(13)$$Dice\;=\;\frac{\left(2*\left|A\cap B\right|\right)}{\left(\left|A\right|+\left|B\right|\right)}=\frac{2*TP}{2*TP+FN+FP}=\frac{2*Precision*Sensitivity}{Precision+Sensitivity}\;$$

The notion || represents the total number of pixels. TP,FN,FP are the true-positive, false-negative, and false-positive prediction rates at the pixel level, respectively.

Precision represents the proportion of correctly predicted positives among all predicted positives. Sensitivity (or Recall) measures the model’s ability to detect actual positives, while Specificity reflects its ability to identify actual negatives. Accuracy captures the overall correctness of the model by accounting for both positive and negative predictions. The Dice coefficient (also known as the F1-score) effectively balances Precision and Sensitivity and is particularly useful in segmentation tasks where both false positives and false negatives are essential. The Jaccard index, also known as Intersection over Union (IoU), measures the intersection between the predicted and ground-truth masks relative to their union.

However, not all metrics are relevant to the nature of a specific segmentation task. Specificity and Accuracy, for example, can appear misleadingly high due to the overwhelming presence of TN, especially when the background dominates the image. In our case, the mean lesion coverage within the lesion-centered 512 × 512 image tiles is approximately 15%, yielding a background-to-lesion pixel ratio of about 7:1. This imbalance inflates both Specificity and Accuracy, leading to an overestimation of segmentation performance. This effect becomes even more pronounced during whole-image inference, where background regions become more prevalent.

Moreover, segmentation evaluation should reflect the clinical context and task-specific challenges. In the case of the detection of AKs, lesions often lack sharply defined borders, with margins that are blurred and gradually blend into the surrounding healthy skin, making precise delineation challenging even for experienced dermatologists. As a result, annotations are inherently approximate. IoU is strict and penalizes any deviation more heavily, making it less suitable for evaluating segmentations with loose annotations. By contrast, the Dice coefficient is less sensitive to boundary deviations, making it more suitable for tasks with imprecise or fuzzy annotations, such as AK segmentation.

Additionally, the segmentations that our experts (IB, GG, DE, AZ) consider acceptable for AK lesions tend to align more closely with the Dice scores, reflecting the metric’s strong agreement with expert judgment (Figure 3). Given its robustness in this context, Dice is adopted throughout the remainder of this study to compare segmentation models in a clinically meaningful way.

To rigorously compare the performance of models, we analyzed the Dice coefficient achieved by each method using non-parametric statistical tests, which are appropriate for repeated measures across the same dataset when necessary. A Friedman test was applied to determine whether performance differences existed across all of the models. Post hoc pairwise comparisons were performed using the Wilcoxon signed-rank test with the Bonferroni correction for multiple comparisons.

Prominent architectures at the crop level were finally evaluated for the detection of AKs in clinical images employing a sliding window approach. This method involves breaking down a large image into smaller overlapping tiles, making predictions on these patches, and then stitching them back together to form the final prediction. This technique enables the model to focus on smaller areas of the image, which is particularly advantageous when working with high-resolution images that display a wide range of textures, color intensities, and lesion appearances. This overlapping strategy helps mitigate the boundary effects that typically arise in segmentation tasks, where regions at the edges of crops may receive less attention from the model, or errors can even occur when the patch boundary splits objects. Once the crops have been processed and predictions are made, the next challenge is to merge these predictions back into a single, coherent segmentation map. Given the overlapping nature of the crops, multiple predictions will be made for the same pixel. Without proper weighting, inconsistencies or artifacts may arise, especially in overlapping regions where predictions from different crops may vary (Figure 4).

To ensure a seamless integration of predictions, we applied a Gaussian weighting technique. The idea is to prioritize the center of each crop when merging the predictions.

Pixels near the center of a crop tend to be more reliable, because they are farther from the crop’s boundaries, where predictions might be affected by edge artifacts or the limited context available to the model.

The Gaussian weight map, $G\left(x,y\right)$, is a 2D map that assigns higher weights to the central pixels and gradually reduces the weights towards the edges. It is defined as follows:(14)$$G\left(x,y\right)\;=\;\exp\left(-0.5\;\times \;\frac{\left(x^{2}\;+\;y^{2}\right)}{\sigma^{2}}\right)$$
where x and y are the pixel coordinates relative to the center of the crop, and σ controls the spread of the Gaussian function. The Gaussian map resembles a bell-shaped curve, where the central region of the crop is emphasized, and the boundary regions are down-weighted.

In our experiments, the Gaussian function’s influence extends up to 1/4th of the crop’s width from its center. This ensures that the center region of each crop, which contains the most confident predictions, is weighted more heavily, while the outer pixels, which may be affected by edge effects, contribute less to the final prediction.

To reconstruct the full segmentation map from the individual crops, the predictions for each crop are multiplied by the corresponding Gaussian weight map, which emphasizes the central pixels and downweighs the predictions near the edges. This weighted prediction is added to a cumulative full-size prediction map.

In addition to the full prediction map, a cumulative weight map is maintained to track the total weight assigned to each pixel during the merging process. This ensures that areas with more overlap receive appropriate normalization.

For each crop, the weighted prediction is added to the full prediction map at its corresponding position, and the Gaussian weight map is added to the cumulative weight map. After all the crops have been processed, the final full-size prediction map is normalized by dividing it by the cumulative weight map, ensuring that each pixel’s final prediction is a weighted average of all the overlapping crops that contributed to it. This process can be expressed as follows:(15)$$Pfinal(x,y)=\frac{\sum_{i}G_{i}(x,y)P_{i}(x,y)}{\sum_{i}G_{i}(x,y)}$$
where $Pfinal\left(x,y\right)$ is the final prediction for pixel $\left(x,y\right)$, and $P_{i}\left(x,y\right)\;and\;G_{i}(x,y)$ are the prediction and the Gaussian weight map for the i-th crop, respectively.

The full-size segmentation map, reconstructed using the Gaussian-weighted predictions, is then thresholded to produce a binary mask.

Besides the overlapping tiles approach, we also experimented with the YOLOv8-Seg model, which processes the entire image to detect and segment AK regions. The YOLOv8-Seg model is an extension of the YOLOv8 architecture explicitly designed for instance segmentation tasks [42,43,44]. While it retains the core object detection framework of YOLOv8, it adds segmentation capabilities, enabling it to predict pixel-wise masks in addition to bounding boxes and class probabilities for objects in an image. This allows the model to not only detect objects but also to separate them from the background with more precise segmentation.

To assess the segmentation performance of the proposed models, we computed the median Dice coefficient (%) for each model.

### 3.4. Implementation Details

The models were implemented in Python 3 using the TensorFlow 2.15 framework or PyTorch 1.11 (HmsU-net, TransU-net). Training was conducted on a system equipped with an NVIDIA A100 GPU and 100 GB of RAM. All models were trained for 100 epochs using the Binary Cross-Entropy loss function and the Adam optimizer, with an initial learning rate of 0.001 and a weight decay of 1 × 10^−6^. The batch size was set to 32, which was chosen based on the memory efficiency and throughput characteristics of the NVIDIA A100 GPU, as recommended by the manufacturer for deep learning workloads.

Due to computational constraints, the image crops were downscaled by a factor of 0.5, resulting in a final resolution of 256 × 256 pixels. For YOLOSegV8 and the whole-image AK segmentation task, the images were instead rescaled to 1120 × 1120 pixels.

## 4. Results

### 4.1. AK Detection in Localized Image Regions (256 × 256 Crops)

To assess the segmentation performance of the proposed models, we computed the median Dice coefficient (%) for each model. Post hoc pairwise comparisons were performed using the Wilcoxon signed-rank test with the Bonferroni correction for multiple comparisons. With 15 pairwise model comparisons, the Bonferroni-adjusted threshold for significance wasα_corrected = 0.05/15 ≈ 0.0033 (16)

The violin plot in Figure 5 visually summarizes these results. Horizontal lines indicate each pairwise test. Asterisks denote adjusted *p*-value thresholds as follows:

*: *p* < 0.01 (nominal, but not significant under Bonferroni);

**: *p* < α_corrected (statistically significant under Bonferroni);

***: *p* < 0.001 (highly significant).

**Figure 5 diagnostics-15-01752-f005:**
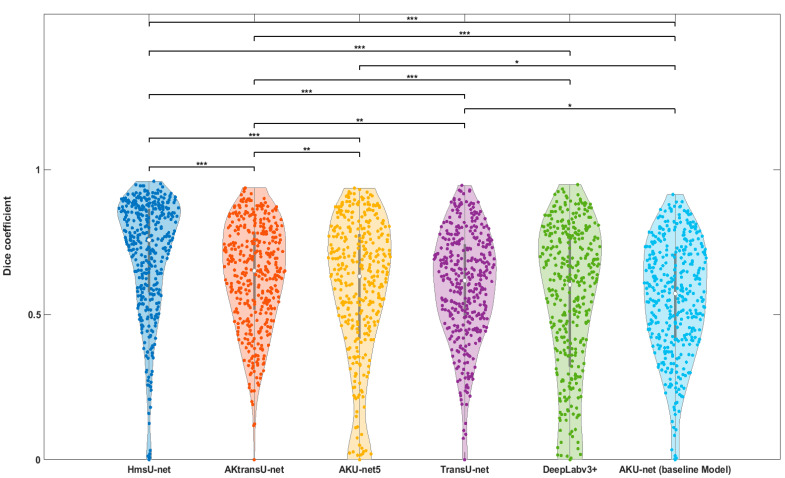
Violin plots of Dice coefficient distributions for six models. Pairwise comparisons were conducted using Wilcoxon signed-rank tests with Bonferroni correction for multiple testing (15 comparisons; α_corrected ≈ 0.0033). Asterisks indicate significance levels: *: *p* < 0.01 (not significant after Bonferroni correction), **: *p* < 0.0033 (Bonferroni-corrected significance), ***: *p* < 0.0 (highly significant). The shape and spread of each violin reflect performance consistency. Models like HmsU-net exhibit a narrow distribution with a high median, indicating both a strong and stable performance across test cases.

Moreover, Table 2 compiles the pairwise statistical comparisons against the baseline AKU-net model [14].

The results indicate that HmsU-net significantly outperforms all other models in terms of both central tendency and consistency of performance. Both HmsU-net and AKtransU-net exhibited highly significant improvements (***) over the reference model, AKU-net, with HmsU-net achieving a substantial increase of more than 18 percentage points in its median Dice score.

In contrast, the improvements observed with TransU-net and AKU-net5, while nominally better than the baseline, did not withstand statistical correction for multiple comparisons and thus lacked statistical robustness. Similarly, DeepLabv3+ failed to show a statistically significant improvement over AKU-net, both in terms of raw performance (60.24% Dice) and statistical significance.

A visual demonstration of the efficiency of the evaluated models is given in Figure 6.

### 4.2. Lesion Detection in Full Clinical Photographs—Whole-Image Inference

To handle lesion detection in whole images, we compared three crop-based scanning strategies during semantic segmentation and evaluated their impact on segmentation performance. The first tested approach was simple block processing, involving non-overlapping image tiling into fixed-size crops (512 × 512 pixels), with predictions directly concatenated. While computationally efficient, this approach introduced artifacts at crop boundaries due to the lack of contextual information. Following, overlapping crop inference was applied using a sliding window with a stride of either 256 × 256 pixels or 128 × 128 pixels, where overlapping predictions were averaged to produce the final segmentation. Finally, we employed a Gaussian-weighted overlapping inference that used the same stride (Table 3). The quantitative evaluation revealed a progressive improvement in segmentation accuracy and boundary continuity across methods. The Gaussian-weighted stride approach achieved the highest segmentation accuracy, with a median Dice score of 65.13%, followed by the overlapping average stride (48.11%) and simple block processing (45.74%). Pairwise comparisons using the Wilcoxon signed-rank test confirmed that both alternative strategies performed significantly worse than the Gaussian-stride baseline (*p* < 0.01 for both comparisons). These results demonstrate that Gaussian weighting over overlapping patches effectively improves segmentation consistency and accuracy in large, high-resolution photographs.

The wide-area segmentation performance of AKtransU-net, using a Gaussian-weighted stride (base model), was compared with that of HmsU-net and YOLOSegV8. The results compiled in Table 4 show that AKtransU-net performed significantly better than HmsU-net at a 256 × 256 stride (*p* < 0.05), while HmsU-net at a 128 × 128 stride and YOLOSegV8 showed no significant difference from AKtransU-net (*p* > 0.05). However, a stride of 128 × 128 pixels requires approximately four times more inference computations than a stride of 256 × 256 pixels for the same image. Figure 7 illustrates the qualitative results of AK detection on full-face images.

We performed an additional evaluation using targeted patch extraction to estimate the upper bound of segmentation performance for each model. Specifically, crops were extracted with their centers aligned to the annotated AK lesions. This lesion-centered extraction strategy ensured that each crop fully encompassed the lesion of interest while minimizing background variability and irrelevant tissue. As a result, the models were evaluated at whole-image detection under optimal conditions that approximate the maximum achievable segmentation performance. These results provide a helpful reference point for interpreting the practical scanning performance of each model under standard whole-image tiling conditions.

Based on the lesion-centered patch extraction, the upper-bound evaluation for whole-image inference confirmed the superiority of hybrid CNN + ViT architectures over their CNN-only counterparts, highlighting the benefit of incorporating global attention mechanisms (Table 5). Moreover, AKtransU-net and HmsU-net achieved the highest segmentation performance. Both models achieved comparable median Dice scores under these ideal conditions (*p* > 0.05), indicating a similar potential when the lesion is fully captured within the input crop. However, the results of the full-image scanning experiments demonstrated substantial performance differences between the models, with AKtransU-net consistently outperforming HmsU-net under standard tiling conditions. This divergence suggests that the scanning strategy and variability in context have a strong influence on models’ behavior. Specifically, the performance of HmsU-net appeared more sensitive to changes in scanning configuration and patch overlap, while AKtransU-net maintained a more consistent accuracy.

These findings underscore the crucial role of scanning methodology in real-world segmentation applications, suggesting that evaluating models solely under optimal conditions may underestimate their variability and robustness during deployment.

## 5. Discussion

The integration of artificial intelligence into clinical photography holds significant potential for enhancing the monitoring of skin conditions, such as actinic keratosis, and for the broader context of skin field cancerization. AK lesions are often subtle, poorly defined, and diffusely distributed across chronically sun-damaged skin, making their accurate and consistent evaluation challenging, even for experienced clinicians. Given that cutaneous field cancerization progresses gradually and exhibits complex spatial patterns, there is a clear need for an advanced semantic segmentation model tailored explicitly to these clinical demands. If AK lesions can be reliably identified across complex, photodamaged skin, then a range of critical capabilities emerge, such as the consistent assessment of disease burden, the early detection of subclinical field changes, and the objective quantification of treatment response over time. Furthermore, such a system would enable standardized follow-up protocols by providing clinicians with detailed, reproducible, and spatially coherent information. In essence, high-performance AK detection is the foundation for longitudinal monitoring, therapeutic planning, and, ultimately, the advancement of the precise and personalized management of field-directed therapies in dermatology.

In this work, we extended a previously introduced U-net architecture enhanced with ConvLSTM modules by further integrating ViT encoding into the skip connections. Our goal was to target improvements where they are most critically needed: preserving spatial detail while incorporating a global contextual understanding. The inclusion of ViT modules in the skip pathways enabled the model to retain fine-grained localization features while simultaneously gaining a broader semantic view of the image. This global awareness is vital for suppressing false positives, as lesion detection benefits from an enhanced understanding of the surrounding tissue context. This is supported by a comparison with the state-of-the-art hybrid architecture HmsU-Net, where our model exhibited a more robust context awareness in whole-image assessments. The achieved Dice score of 65.13% represents the highest performance reported to date, demonstrating significant progress (Table 4).

To mitigate boundary artifacts and improve the smoothness and coherence of the final segmentation map, Gaussian stride processing was employed, which applies a weighted blending of overlapping tile outputs and enhances detection accuracy by approximately 17% (Table 3). While this method enables the processing of full-size images on limited hardware, it inherently fragments spatial information. Moreover, upper-bound evaluations revealed the gap between the tile-wise and true detection accuracy of targeted lesions, indicating room for improvement (Table 5).

Alternatively, YOLOv8-Seg offers a segmentation solution that processes the entire image in a single pass, preserving the global context. While its performance is comparable to that of the proposed AKTransU-net (Table 4), it appears to have reached a performance plateau in the context of AK segmentation. A key limitation is its maximum input resolution of 1120 × 1120 pixels, which is constrained by hardware and the architectural design. This restricts its capacity for further improvement in high-resolution dermatological image analysis.

In contrast, tile-based processing offers a more flexible and scalable framework, with significant potential for further improvement through advancements in both model architecture and tile management strategies.

Finally, while the current architecture prioritizes segmentation accuracy and contextual understanding, its computational complexity remains a limitation, particularly due to the inclusion of multi-scale Transformer blocks. In future work, we aim to explore model compression techniques, lightweight Transformer variants, and efficient attention mechanisms to improve its inference speed and make the model more suitable for deployment on resource-constrained devices.

## 6. Conclusions

The introduction of Transformers into visual tasks has significantly expanded the deep learning armamentarium, providing powerful tools for modeling long-range dependencies and capturing a global context—capabilities that are particularly valuable in clinical imaging. This is especially important for challenging applications, such as skin field cancerization, where the detection of AK lesions depends on both fine detail and a broader contextual understanding. As high-resolution imaging becomes increasingly central to dermatologic care, Transformer-driven context modeling offers a promising path forward for robust, scalable, and clinically meaningful AI-assisted monitoring.

## Figures and Tables

**Figure 1 diagnostics-15-01752-f001:**
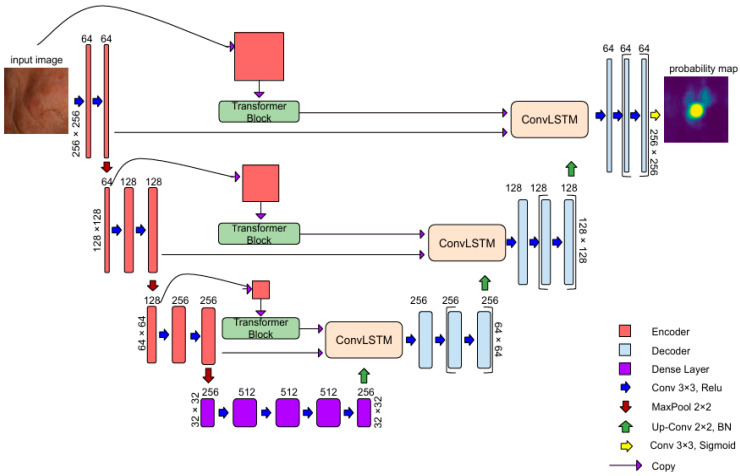
AKtransU-net overview. The encoder consists of four stages, where feature maps are progressively down-sampled using convolutional and pooling layers. Each encoder-level input is passed to a Transformer block. ConvLSTM units in skip connections sequentially process convolutional, Transformer, and up-sampled decoder features.

**Figure 2 diagnostics-15-01752-f002:**
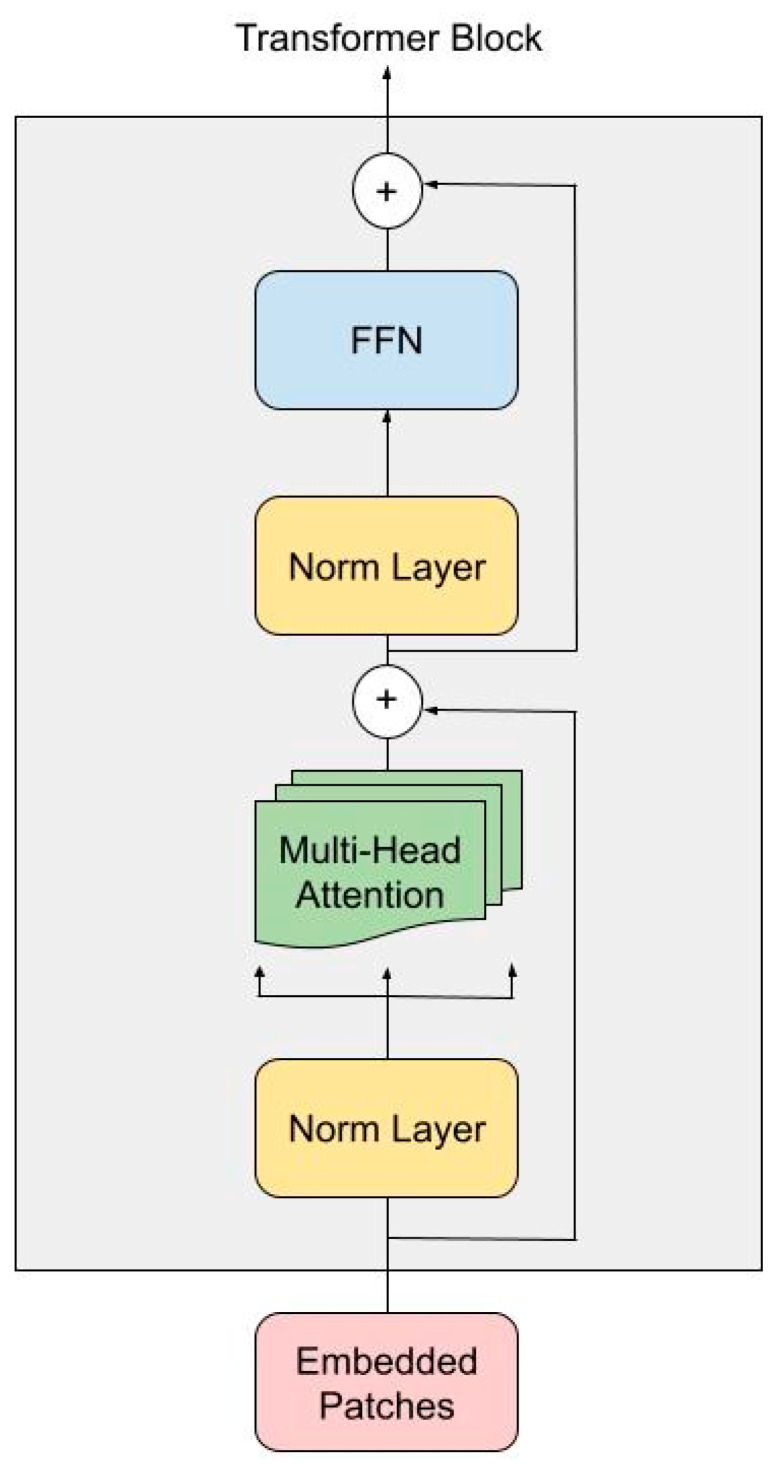
ViT encoder adapted from [15].

**Figure 3 diagnostics-15-01752-f003:**
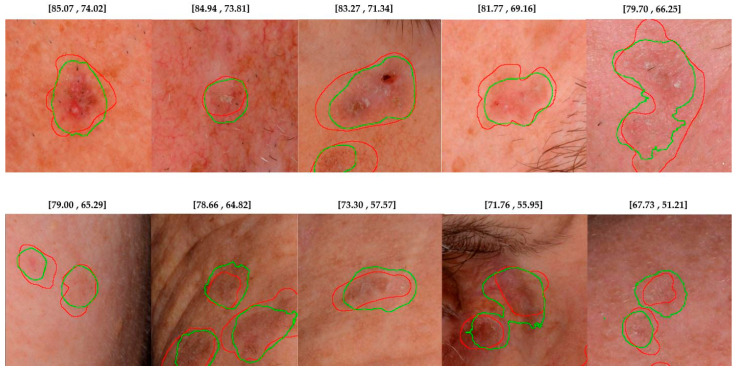
Segmentation results considered acceptable by clinical experts, with corresponding Dice and Jaccard scores [Dice, Jaccard]. Red contours represent ground-truth annotations, while green contours show model predictions. In cases where segmentations exhibit slight boundary mismatches, the Dice scores remain relatively high, whereas the Jaccard scores drop more sharply due to their stricter penalization of disagreement. Dice better aligns with clinical judgment in cases of ambiguous lesion margins, such as those seen in AK, where exact boundaries are often difficult to define.

**Figure 4 diagnostics-15-01752-f004:**
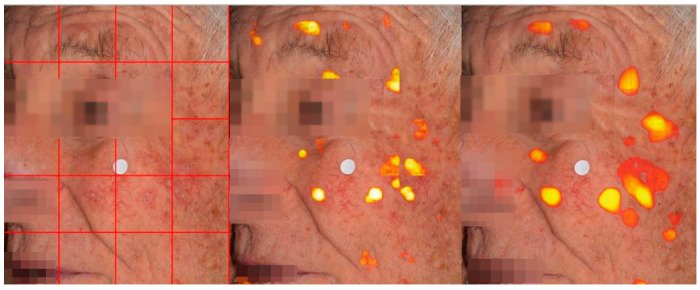
The left panel shows the original facial image overlaid with a grid indicating tile-based segmentation. The middle panel illustrates the result of simple tile-wise processing followed by direct stitching, which leads to missed detections and boundary artifacts in AK localization. In contrast, the right panel demonstrates the improved AK detection achieved through Gaussian-weighted blending across tile boundaries, which alleviates discontinuities and enhances detection’s accuracy and consistency.

**Figure 6 diagnostics-15-01752-f006:**
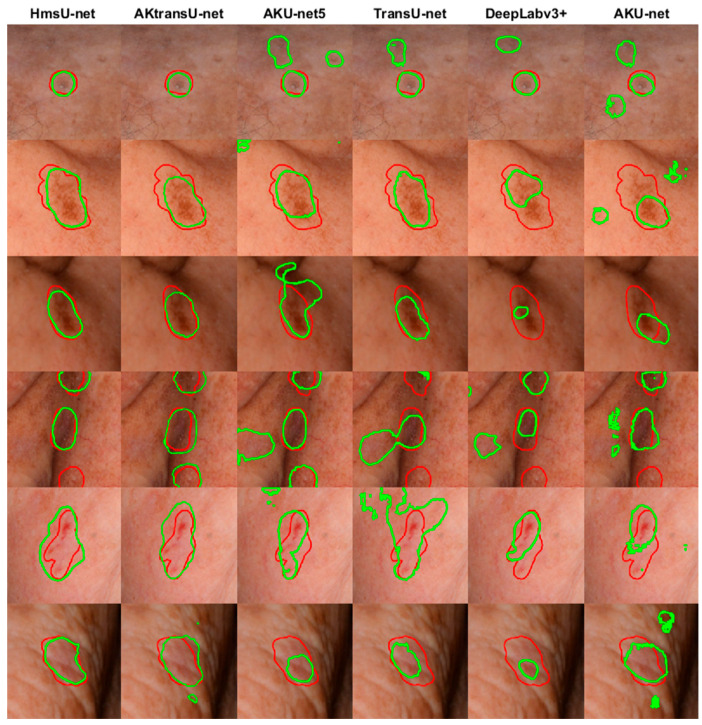
Qualitative comparison of lesion segmentation results on cropped cases where the base model, AKU-net, underperformed. Red contours indicate the ground-truth segmentations, while green contours represent the predicted segmentations by each model.

**Figure 7 diagnostics-15-01752-f007:**
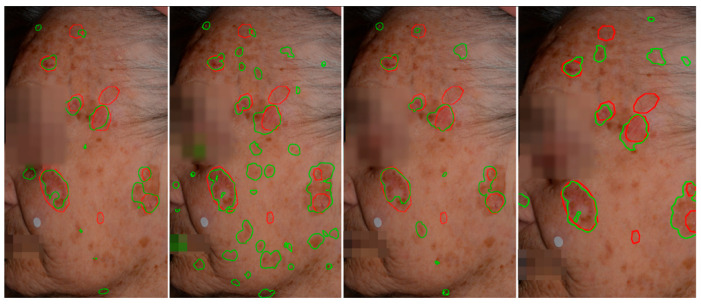
Qualitative comparison of AK detection results on whole-face images. From left to right: AKtransU-net (stride, 256 × 256), HmsU-net (256 × 256), HmsU-net (128 × 128), and YOLOSegV8. Red contours denote the ground-truth segmentations, while green contours indicate the predictions generated by each model.

**Table 1 diagnostics-15-01752-t001:** Dataset splitting into train, validation, and test sets.

	Patients	Images	Image Crops	Augmentation Technique
Train	83	410	13,190	Translation
Validation	15	100	3298	Translation
Test	17	59	403	None
Total	115	569	16,891	

**Table 2 diagnostics-15-01752-t002:** Median Dice coefficients and statistical significance (*p*-values) compared to the baseline AKU-net model. *: *p* < 0.01 (nominal, but not significant under Bonferroni), ***: *p* < 0.001 (highly significant).

Model	Median Dice (%)	*p*-Value (vs. Baseline Model)
AKU-net (baseline model)	57.21	
AKU-net5	63.16	*
DeepLabv3+	60.24	-
TransU-net	61.70	*
HmsU-net	75.68	***
AKtransU-net	65.10	***

**Table 3 diagnostics-15-01752-t003:** Performance of the whole-image AKtransU-net. Pairwise comparisons using the Wilcoxon signed-rank test showed that Gaussian weighting (the base method) significantly improved segmentation consistency compared to alternative strategies, such as block processing or uniform averaging of overlapping patches. Decreasing the stride from 256 to 128 pixels did not improve the segmentation accuracy of the AKtransU-net model.

Scanning Method	Median Dice (%)	*p*-Value (vs. Baseline Method)
Gaussian-weighted stride (256 × 256) baseline	65.13	—
Gaussian-weighted stride (128 × 128)	64.70	*p* > 0.05
Average overlapping stride (256 × 256)	48.11	*p* < 0.01
Non-overlapping block processing	45.74	*p* < 0.01

**Table 4 diagnostics-15-01752-t004:** Comparison of segmentation performance between AKtransU-net, HmsU-net, and YOLOSegV8 models. AKtransU-net with a stride of 256 × 256 pixels served as the baseline model. Pairwise comparisons were performed using the Wilcoxon signed-rank test.

Model	Median Dice (%)	*p*-Value (vs. Baseline Model)
AKtransU-net stride (256 × 256) baseline	65.13	—
HmsU-net (256 × 256)	52.39	*p* < 0.05
HmsU-net (128 × 128)	61.21	*p* > 0.05
YOLOSegV8	61.44	*p* > 0.05

**Table 5 diagnostics-15-01752-t005:** Upper-bound segmentation performance confirmed the superiority of hybrid CNN + ViT architectures (AKtransU-net and HmsU-net) over their CNN-only counterparts.

Model	Median Dice (%)
AKU-net (base model)	59.21
AKU-net5	61.35
DeepLabv3+	60.70
TransU-net	67.51
AKtransU-net	73.34
HmsU-net	74.02

## Data Availability

The code is available upon request.

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
