# Peer review of "AΚtransU-Net: Transformer-Equipped U-Net Model for Improved Actinic Keratosis Detection in Clinical Photography"

_diagnostics, 2025, doi:10.3390/diagnostics15141752_

Round 1

Reviewer 1 Report

Comments and Suggestions for Authors
  1. Highlight the purpose of incorporating UNET in the abstract.
  2. Check for repetitive citations on page 2 (Reference 3 is cited in lines 49 and 56).
  3. In Section 2, the evolution of transformers in medical image segmentation and the importance of skip connections are discussed. However, this content is irrelevant to this section and should be included in a separate section dedicated to related works.
  4. On page 8, line 246, the sigma symbol needs to be included.
  5. Provide justification for using a small dataset with the complex architecture.
  6. Explain why transformers are used and include the ablation study's findings comparing performance with and without transformers. Add a separate section for the proposed methodology.
  7. On page 12, Figure 5 is cited but not included in the manuscript. Please verify the sentence.

Reviewer 2 Report

Comments and Suggestions for Authors

The study presents the AKtransU-net architecture, which integrates Vision Transformer blocks into a multi-scale ConvLSTM-supported U-Net structure for the segmentation of actinic keratosis (AK) lesions from clinical photographs. The study contributes to the literature by presenting comparisons with existing CNN-based and hybrid architectures, a Gaussian-weighted tiling strategy for lossless full-image extraction on high-resolution facial photographs, and statistical analysis.
The parts that need correction are as follows;
• A more detailed explanation of the dataset would be useful.
• Details such as data augmentation techniques, hyperparameter tuning, or early stopping are missing in the training parameters.
• Figure numbers should be checked. There is an error.
• A comparison is made with the YOLOv8-Seg model, but it is not sufficiently discussed why this model was chosen or which features are suitable for AK segmentation.
• Only AKtransU-net and HmsU-net are compared in the upper bound evaluation. The comparison would be more comprehensive if the performance of other models (e.g., TransU-net, DeepLabv3+) in this scenario were also presented.

• The paper highlights the superiority of AKtransU-net over HmsU-net, but the clinical implications of this superiority are not sufficiently discussed. For example, how meaningful is the Dice score of 65.13% in the clinical decision-making process? How does this improve dermatologists’ current methods?

• It is stated that the addition of Transformer blocks increases the computational complexity of the model, but the practical implications of this complexity (e.g., inference time, hardware requirements) are not quantified.

Reviewer 3 Report

Comments and Suggestions for Authors

The paper proposes AKTransU-net, a hybrid U-Net-based structure aimed at improving spatial detail preservation and global contextual comprehension. The model integrates ViT-based Transformer blocks at various encoding levels to enhance feature representations, subsequently transmitting them through ConvLSTM modules embedded in the skip connections. This arrangement enables the network to preserve semantic coherence and spatial continuity during the segmentation process. The topic of the paper is interesting, however, some comments need to be addressed. The paper lacks a related work section. The novelty and contributions are vague.

.

The abstract

Kindly specify whether the dataset utilised in this study is proprietary or a publicly accessible benchmark dataset.

Please add numerical findings.

Please combine the paragraphs of the abstract into one paragraph around 250 words.

Introduction:

The abbreviation of the convolutional neural network is CNN, not AKCNN.

What is AK burden? It is repeated all over the manuscript. I think it needs revision.

The originality of the paper is ambiguous. Kindly elucidate the originality in relation to prior research.

Kindly delineate the innovations and contributions in bullet points.

Please dedicate a section to related work and literature review.

Materials and Methods

Please revise abbreviations, they should be define only once

A wide range of deep learning techniques are available for segmentation. What was the rationale for employing U-Net?

Moreover, multiple iterations of U-Nets exist. Kindly specify the version.

Please provide a rationale for selecting ViT in light of the availability of various deep learning models.

Figure 1 resolution is poor.

Kindly specify the criteria for inclusion and exclusion in the data collection process.

Kindly incorporate samples into the images of the dataset.

What hyperparameters are used for deep learning models in the described system?

Experimental Results

Please add more performance metrics to the results like IoU , accuracy, sensitivity, and specificity.

I can see that HmsU-net achieved higher results than AKtransU-net, please explain.

Discussions

What are the limitations and future directions of the presented study?

Comments on the Quality of English Language

Some awkward phrases appear. it seems like the text has been AI-generated and paraphrased in some paragraphs

Round 2

Reviewer 1 Report

Comments and Suggestions for Authors

Change the column name 'crops' in Table 1 to a relevant name for the data augmentation technique.

Author Response

Comment: Change the column name 'crops' in Table 1 to a relevant name for the data augmentation technique.

Response: We thank the reviewer for the valuable comment. We would like to clarify that the column in question refers to the number of image crops used during training, validation, and testing. For the test set, we used an independent set of patient images, from which only center-lesion crops were extracted—no data augmentation was applied in this case. To better reflect this, we have renamed the column to "Image Crops". Additionally, the third column in the table has been labeled with the relevant data augmentation technique used during training. Please refer to lines 323–328 in the revised manuscript.

Reviewer 3 Report

Comments and Suggestions for Authors

The authors have addressed my comments

Author Response

Thank you very much for your insightful revision.